# Extraction of Manganese and Iron from a Refractory Coarse Manganese Concentrate

Junhui Xiao [1,2,3,4,5,6,*], Kai Zou [1,4], Tao Chen [1,4], Wenliang Xiong [1,2,3] and Bing Deng [1,2,3]

1    Sichuan Provincial Engineering Lab of NonMetallic Mineral Powder Modification and High-Value Utilization, Southwest University of Science and Technology, Mianyang 621010, China; zoukaiswust@163.com (K.Z.); chentao@mails.swust.edu.cn (T.C.); wenliangxiong2020@163.com (W.X.); dbing86@163.com (B.D.)
2    Dongfang Boiler Group Co., Ltd., Zigong 643001, China
3    Institute of Multipurpose Utilization of Mineral Resources, Chinese Academy of Geological Sciences, Chengdu 610041, China
4    Guangdong Provincial Key Laboratory of Radioactive and Rare Resource Utilization, Guangdong Institute of Mineral Resources Application, Shaoguan 512026, China
5    Key Laboratory of Sichuan Province for Comprehensive Utilization of Vanadium and Titanium Resources, Panzhihua University, Panzhihua 617000, China
6    Key Laboratory of Ministry of Education for Solid Waste Treatment and Resource Recycle, Southwest University of Science and Technology, Mianyang 621010, China
*    Correspondence: xiaojunhui@swust.edu.cn; Tel.: +86-1399-019-0544

**Abstract:** In this research, the coarse manganese concentrate was collected from a manganese ore concentrator in Tongren of China, and the contents of manganese and iron in coarse manganese concentrate were 28.63% and 18.65%, respectively. The majority of the minerals in coarse manganese concentrate occur in rhodochrosite, limonite, quartz, olivine, etc. Calcium chloride, calcium hypochlorite, coke, and coarse manganese concentrate were placed in a roasting furnace to conduct segregation roasting, which resulted in a partial chlorination reaction of iron to produce $FeCl_3$, ferric chloride reduced to metallic iron and adsorbed onto the coke, and rhodochrosite broken down into manganese oxide. Iron was extracted from the roasted ore using low-intensity magnetic separation, and manganese was further extracted from the low-intensity magnetic separation tailings by high-intensity magnetic separation. The test results showed that iron concentrate with an iron grade of 78.63% and iron recovery of 83.60%, and manganese concentrate with a manganese grade of 54.04% and manganese recovery of 94.82% were obtained under the following optimal conditions: roasting temperature of 1273 K, roasting time of 60 min, calcium chloride dosage of 10%, calcium hypochlorite dosage of 5%, coke dosage of 10%, coke size of −1 mm, grinding fineness of −0.06 mm occupying 90%, low-intensity magnetic field intensity of 0.14 T, and high-intensity magnetic field intensity of 0.65 T. Most minerals in the iron concentrate were Fe, $Fe_3O_4$, and a small amount of $SiO_2$ and $CaSiO_3$; the main minerals in the manganese were MnO, and a small amount of $Fe_3O_4$, $SiO_2$, and $CaSiO_3$. The thermodynamic calculation results are in good agreement with the test results.

**Keywords:** manganese; iron; rhodochrosite; roasting; magnetic separation

## 1. Introduction

Manganese is one of the most widely used metals in modern industry. Because of its various valence and excellent physical and chemical properties, manganese is widely used in steel, metallurgy, machinery, environmental protection, fire protection, and other fields. It plays a pivotal role in promoting industrial development and economic progress. China is rich in manganese ore resources, but the average manganese grade is only 21.4%, of which the rich manganese ore (manganese oxide ore, Mn grade >30%, manganese carbonate ore, Mn grade >25%) is only 6.4% and the Mn grade >48% (international commodity grade) is almost zero, and the poor manganese ore accounts for 93.6%. Manganese ore composition is complex, with high contents of phosphorus, iron, and silicon, and is often associated

with silver, lead, zinc, and so on. According to statistics, in the manganese ore deposits of China, phosphorus content exceeds the standard, $\omega(P) \leq 0.003\%$ accounts for 49.6%; iron content exceeds the standard, (Mn/Fe $\geq$ 6) accounts for 73%; and silicon content exceeds the standard, $\omega(SiO_2) > 10\%$ accounts for 68% [1–4].

At present, the process of manganese carbonate resources mainly includes ore washing, gravity separation, magnetic separation, flotation, leaching, and combined process. (1) Ore washing is often used as a pretreatment method for mineral separation, which can improve the grade of selected ore. According to the characteristics of selected ores, the appropriate ore washing process can be selected to optimize the subsequent grinding process, including removing the argillization minerals and alleviating equipment blockage [4]. (2) The gravity separation process is a traditional ore dressing process, which has the advantages of low cost, good desliming effect, wide application range, and small amounts of environmental pollution. In the separation of manganese carbonate ore, a good separation effect can be achieved for the selected ore. Reconcentration is often used as a pre-enrichment method to reduce the cost of subsequent processes [5]. (3) The magnetic separation process requires more advanced equipment, so the investment cost is high, but it has the advantages of simple operation, environmental friendliness, and low subsequent operating costs. The concentration of manganese carbonate obtained by magnetic separation is higher than that of heavy concentrate, which is an important means of separating modern industrial manganese ore resources. With the complexity of selected ores, the research and development of an efficient magnetic separator should be strengthened [6,7]. (4) The flotation process can achieve a good separation effect and relatively high concentration in the treatment of low-grade manganese carbonate ore with fine grain size, which is one of the important ways of treating this kind of ore. However, due to the complex composition of ores, the requirement for flotation reagents is also higher; therefore, to effectively use flotation technology to recover manganese resources, the development of efficient flotation reagents is the only way [8]. (5) The extraction process to recover manganese resources in manganese carbonate has a good mineral processing index, and can effectively realize the recovery of useful associated minerals. Compared with the traditional beneficiation process, the leaching process index is better, but the unit time processing capacity is low, the beneficiation cost is high, and the environmental impact is great. The leaching process can achieve good results and has strong pertinence in treating some specific ores. The research on leaching technology should be improved. The latest microbiological leaching and electrodeposition leaching are still in the experimental stage and have few industrial applications due to their harsh environmental requirements. (6) Combined processes have a variety of complementary advantages; it can improve the comprehensive separation index and the recycling of associated mineral products, maximize resource efficiency, and technologically and economically solve the difficult problems of manganese carbonate ore dressing, making up for the deficiency of single process while optimizing the separation process [9–11].

Carbonate manganese ores are mainly composed of rhodochrosite ($MnCO_3$), magnesium rhodochrosite (($Mn, Mg)CO_3$), kutnahorite(($Ca, Mg, Mn)CO_3)_2$), calcite ($CaCO_3$), and siderite ($FeCO_3$). These ores contain silicate, sulfur, and iron impurities. These ores show relatively complex and fine mineral associations which make conventional beneficiation processes a bit difficult. These ores are calcined before subsequent processing. Manganese carbonates ores are relatively different from other two types of ores. These ores contain very high Loss on Ignition ($CO_2$ losses) which impact the yield of the product during pyrometallurgical processing. Calcination of these ores cause $CO_2$ emission which is undesired for environmental point of view. These ores are relatively easy to grind and leach, however the liberation size is very fine. The undesired impurities can be removed by gravity or magnetic separation process and subsequently these can be blended and used for ferroalloy making. Flotation and hydrometallurgical processing methods shown positive results, however these are still not very useful for upgradation of very low-grade ores. Although these ores need extra energy during smelting-reduction mainly due to

endothermic burning of carbonates, these ores are more suitable for production manganese compounds considering their physical and chemical characteristics. In addition, strengthening the recovery of associated valuable metals will be beneficial to the development of manganese carbonate ore resources [12–14].

Manganese carbonate ore deposits in Tongren, China are mostly fine or fine-grained inlaying, with high iron, calcium, and magnesium carbonate content. The inlaying relationship between manganese ore and other gangue minerals is complex, and there are many kinds of minerals, so the ore is easy to argillite [15,16]. Due to the low grade of manganese in manganese resources, the useful mineral mosaic granularity is relatively fine, especially the close symbiotic relationship between iron and manganese, which leads to high production cost and low separation efficiency. Currently, local enterprises use carbonate manganese ore as raw material: the coarse jaw crusher is broken after reaching a certain size of fragment, after breaking the fine grain of manganese carbonate by dressing the high grade manganese ore beneficiation workshop. Then, it is dried. Finally, it is placed into a grinding machine to grind it into a particle size of less than 0.154 mm of qualified products, and a grain size greater than 0.154 mm of unqualified products, to move qualified manganese carbonate powder into the bunker to use or into the combination barrel fluid, as raw materials for the production of electrolytic manganese. Due to the high content of iron and the high content of iron particles in the electrolyte, it has a negative effect on the production of electrolytic manganese. Therefore, it is necessary to extract iron and manganese and obtain qualified concentrate products from the coarse manganese concentrate and remove impurities of iron concentrate and manganese concentrate, and it will promote the development and utilization of the local manganese ore resources.

## 2. Materials and Methods

### 2.1. Sampling

The coarse manganese concentrate with Mn grade of 28.63% and Fe grade of 18.65% was obtained from a high iron-bearing manganese carbonate ore in Tongren, China by spiral chute gravity separation, and the material size was less than 0.154 mm occupying 90%. Quartz, crystal, or fine granular Cain, about 0.001–0.005 mm in size, with occasional 0.01–0.02 mm particles. Calcite, feldspar, and mica have more crumb, fine vein aggregation distribution, a small amount of fine dispersed-shaped distribution of carbonaceous organic matter and rhodochrosite. The late quartz veinlet wear cut, mica, small scales, calcite, fine-grained, and quartz have more of a crumb aggregation distribution, with three levels of about 35%. There are two occurrence states of limonite: one is self-shaped and semi-self-shaped granular, with a size of 0.05–0.3 mm and scattered distribution; the other has a fine disseminated distribution, with a size of about 0.001–0.004 mm, and it is occasionally seen in grass mildew-shaped fine grain aggregate, with a content of about 5%. The main chemical composition analysis of the manganese concentrate is shown in Table 1.

**Table 1.** Main chemical composition analysis of the coarse manganese concentrate (%).

| Mn | Fe | C | S | P | As | $Al_2O_3$ | $SiO_2$ | MgO | CaO | $Na_2O$ |
|-------|-------|------|------|------|-------|-----------|---------|------|------|---------|
| 28.63 | 18.65 | 6.88 | 0.15 | 0.23 | 0.005 | 5.36 | 24.34 | 3.93 | 3.27 | 0.52 |

### 2.2. Main Chemical Reagent and Equipment

The main chemical reagents used in this experiment are calcium chloride (purity, 99.5%) and calcium hypochlorite (purity, 99.5%), both of which come from Beijing Chemical Reagents Research Institute Co. Ltd., Beijing, China. The coke from Shanxi Coking Coal Group Co. Ltd. (Taiyuan, China) was crushed and divided into five particle sizes of −2.5 mm, −2 mm, −1.5 mm, −1 mm, and −0.5 mm as reducing agents in the process of segregation roasting. The entire quality analysis of coke is shown in Table 2.

**Table 2.** Entire quality analysis of coke ($M_{ad}$: air drying base content; $A_d$: ash content; $V_{daf}$: volatile content; $FC_d$: fixed carbon content).

| $M_{ad}$ (%) | $A_d$ (%) | $FC_d$ (%) | $V_{daf}$ (%) | Characteristic of Char Residue |
|---|---|---|---|---|
| 0.45 | 1.68 | 97.12 | 1.06 | 2 |

The main equipment used in the experiment are: roasting atmosphere furnace ($\leq 1300\,^{\circ}C$, Shanghai Shiyan Electric Furnace Co., Ltd. Shanghai, China), electromagnetic wet drum magnetic separator (CRS-400 × 300, Jilin Exploration Machinery Factory, Changchun, China), cone ball mill ($\Phi$ 240 × 90 mm, Jilin Exploration Machinery Factory, Changchun, China), disc grinder ($\Phi$ 300 × 150 mm, Jilin Exploration Machinery Factory, Changchun, China), drying box (Shanghai Shiyan Electric Furnace Co., Ltd., Shanghai, China), vacuum filter ($\Phi$ 300, Southwest Chengdu Experimental Equipment Co., Ltd., Chengdu, China), and ceramic mortar ($\Phi$ 100, Beijing Grinder Instrument Co., Ltd., Beijing, China).

*2.3. Procedure Design*

The coarse manganese concentrate (a mass of 100 g for each roasting test), the chlorinating agent (4–12%), reducing agent (6–14%), and additives (1–9%) were mixed and put into the roasting furnace. These mixtures were heated to a certain temperature (1123–1323 K) under a reducing atmosphere, where the chlorinating agent was used to form volatile metal chlorides and additives were used to enhance the chlorination reaction. For each magnetic test, segregation roasting ores were put into a 6.25 $dm^3$ $\Phi$ 240 × 90 conical ball mill. The grinding density was set at 66.67%. The pulp was then placed in an XCGS-13 ($\Phi$ 50, overall dimensions 1000 mm × 800 mm × 500 mm) Davis Magnetic Tube (Jilin Exploration Machinery Plant, Shenyang, China) with a special magnetic field intensity. The low-intensity magnetic product was used for the iron concentrate, and the non-magnetic product (low-intensity magnetic separation tailings) was placed in a SLon-100 high gradient magnetic separator ($\Phi$ 100 mm × 100 mm, overall dimensions SLon-100 mm, Ganzhou Institute of Nonferrous Metallurgy, Ganzhou, China) with a special magnetic field intensity. The high-intensity magnetic separation product was the manganese concentrate. The products were filtered, dried, weighed, sampled, and tested for further evaluation based on the grade and recovery of manganese and iron in iron concentrate and manganese concentrate, respectively. The calculation formulas of iron recovery and manganese recovery are shown in Equations (1) and (2).

$$\text{Iron recovery} = Q_1 \times \beta_1 / (Q_1 \times \beta_1 + Q_2 \times \beta_2)\, 100\% \tag{1}$$

$$\text{Manganese recovery} = Q_3 \times \beta_3 / (Q_3 \times \beta_3 + Q_4 \times \beta_4)\, 100\%, \tag{2}$$

where $Q_1$ is the weight of iron concentrate, g; $Q_2$ is the weight of low-intensity magnetic separation tailings, g; $Q_3$ is the weight of manganese concentrate, g; $Q_4$ is the weight of high-intensity magnetic separation tailings, g; $\beta_1$ is the iron grade of iron concentrate, %; $\beta_2$ is the iron grade of low-intensity magnetic separation tailings, %; $\beta_3$ is the manganese grade of manganese concentrate, %; and $\beta_4$ is the manganese grade of high-intensity magnetic separation tailings, %.

*2.4. Analysis and Characterization*

The chemical composition of solid materials (including coarse manganese concentrate, roasted ore, iron concentrate, and manganese concentrate) was analyzed by a Z–2000 atomic absorption spectrophotometer (Hitachi Co., Ltd., Tokyo, Japan). The mineral phase composition of solid substances (including coarse manganese concentrate, roasted ore, iron concentrate, and manganese concentrate) was analyzed by X-ray diffraction (X Pert Pro, Panaco, The Netherlands). The microstructure of the solid products (including coarse manganese concentrate, roasted ore, iron concentrate, and manganese concentrate) was observed by SEM (S440, Hirschmann Laborgerate GmbH & Co. KG, Eberstadt, Germany)

equipped with an energy dispersive X-ray spectroscopy (EDS) detector (UItra55, Carlzeiss-NTS GmbH, Hirschmann Laborgerate GmbH & Co. KG, Eberstadt, Germany).

## 3. Results and Discussion

### 3.1. Mineral Composition Analysis of Coarse Manganese Concentrate

The X-ray diffraction (XRD) analysis of the coarse manganese concentrate is shown in Figure 1. The main minerals in coarse manganese concentrate include rhodochrosite, limonite, clay mineral, carbon organic matter, and quartz. The clay minerals are mainly hydromica, which takes the form of fine scales and needles, and the carbonaceous organic matter takes the form of dust. The aggregation takes the form of banding, indicating the banded structure of the ore. Pellet, agglomerate rhodochrosite, agglomerate, vein-like quartz Muscovite, and limonite are embedded in and between the banding, and the content is about 55%. Rhodochrosite, with an implicit crystalline or fine granular size of about 0.0005–0.002 mm, with a mud crystal structure, takes a spherulitic, massive collection, a small amount of fine dispersed-shaped distribution of clay minerals and quartz mica in the aggregate. The spherulitic, massive aggregate size is about 0.05–0.030 mm, with fine grained quartz inside, scaly white mica, inclusions, an even distribution of dust carbon and limonite. The local spherulitic, massive collection has a zonal distribution, and the content is about 15%.

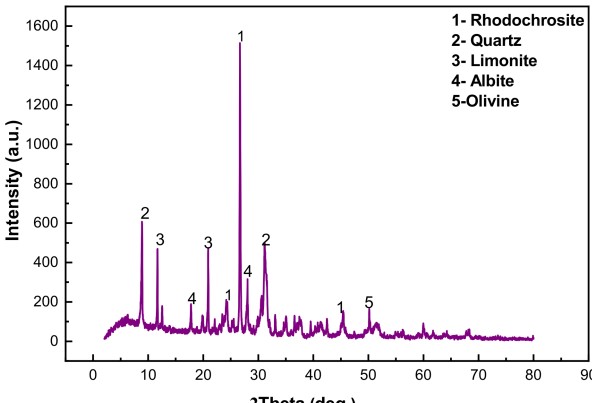

**Figure 1.** X-ray diffractogram (XRD) of the coarse manganese concentrate.

### 3.2. Thermodynamic Analysis of Limonite and Rhodochrosite

Temperature is one of the most important factors for a solid-gas reaction. Temperature can alter the thermodynamic potential of a reaction. After the limonite is decomposed by heat, the crystal water is removed to form hematite. The possible reactions of hematite in the reductive atmosphere of C, CO, and $H_2$ are shown in Equations (3)–(10). Rhodochrosite decomposes into manganese oxide and carbon dioxide when heated, and the possible reactions between MnO and $H_2$, CO, C are shown in Equations (11)–(14).

Limonite and rhodochrosite all belong to weak magnetic minerals, and metallic iron and magnetite have strong magnetic properties and can be recovered by low intensity magnetic separation. Figure 2a shows that iron ore reduction temperature should be higher than 873 K. The results in Figure 2b reveal that when the temperature is 273–1473 K, C, CO, and $H_2$ cannot reduce manganese oxide to manganese metal. Therefore, raising the transformation rate of metallic iron and magnetite is favorable for separation of iron and manganese [17–19]. In this study, chlorine salt was added in the roasting process as an auxiliary agent. Chlorine salt would produce hydrogen chloride gas in the roasting process. Different from the single reduction roasting and chlorination roasting, the effect of different roasting conditions on the separation and extraction of iron and manganese was needed to obtain the ideal process parameters.

$$2CO_{(g)} = C + CO_{2(g)} \tag{3}$$

$$C + 6Fe_2O_3 = 4Fe_3O_4 + CO_{2(g)} \tag{4}$$

$$Fe_3O_4 + C = 3FeO + CO_{(g)} \tag{5}$$

$$C + FeO = Fe + CO_{(g)} \tag{6}$$

$$H_{2(g)} + 3Fe_2O_3 = 2Fe_3O_4 + H_2O_{(g)} \tag{7}$$

$$CO_{(g)} + 3Fe_2O_3 = 2Fe_3O_4 + CO_{2(g)} \tag{8}$$

$$3H_{2(g)} + Fe_2O_3 = 2Fe + 3H_2O_{(g)} \tag{9}$$

$$3CO_{(g)} + Fe_2O_3 = 2Fe + 3CO_{2(g)} \tag{10}$$

$$MnCO_3 = MnO + CO_{2(g)} \tag{11}$$

$$C + MnO = Mn + CO_{(g)} \tag{12}$$

$$CO_{(g)} + MnO = Mn + CO_{2(g)} \tag{13}$$

$$H_{2(g)} + MnO = Mn + H_2O_{(g)} \tag{14}$$

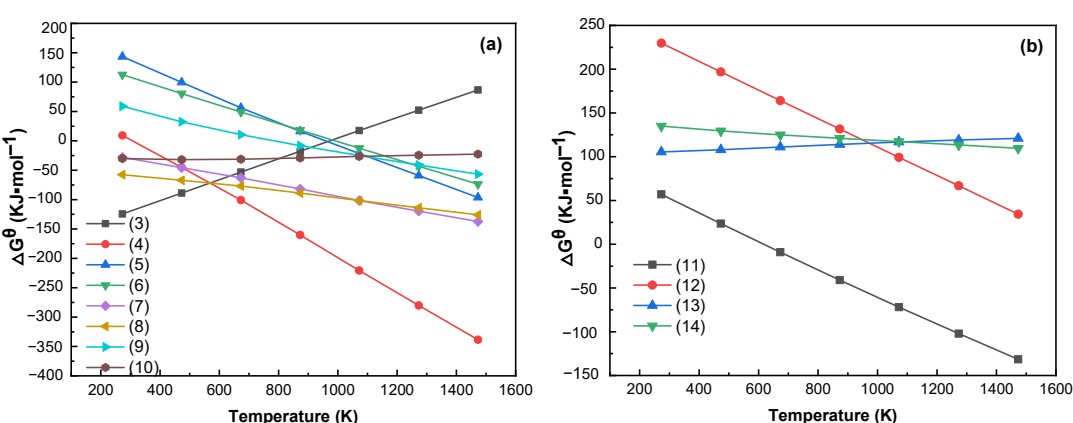

**Figure 2.** Correlation of standard Gibbs free energy (ΔGθ) with temperature for Reactions (**a**) Reactions (3)–(10); (**b**) Reactions (11)–(14).

### 3.3. Iron Extraction from Coarse Manganese Concentrate

#### 3.3.1. Effect of Roasting Temperature

Various roasting temperature tests were carried out under the following conditions: roasting time of 45 min, calcium chloride dosage of 8%, coke dosage of 8%, coke size of −1.5 mm, magnetic field intensity of 0.1 T, and grinding fineness of −0.08 mm occupying 90%. It can be seen from Figure 3a that when the roasting temperature is lower than 1273 K, the iron grade of the iron concentrate decreases, whereas when the roasting temperature is higher than 1273 K, the iron grade of the iron concentrate increases. The recovery of iron concentrate decreases with the increase of temperature. This indicates that increasing the roasting temperature is beneficial to improving the iron grade of the iron concentrate, but the high temperature will lead to a decrease in iron recovery [19,20]. Therefore, the roasting temperature of 1273 K is relatively ideal, and an iron concentrate with iron grade of 62.22% and iron recovery of 64.99% can be obtained.

#### 3.3.2. Effect of Roasting Time

To obtain a suitable roasting time, tests with different roasting times were carried out under the following conditions: roasting temperature of 1273 K, calcium chloride dosage of 8%, coke dosage of 8%, coke size of −1.5 mm, magnetic field intensity of 0.1 T, and grinding fineness of −0.08 mm occupying 90%. Figure 3b shows that a prolonged roasting time increases the iron recovery of iron concentrate, but a longer roasting time reduces the iron grade. When the roasting time was increased to 60 min, the iron grade and iron

recovery increased to 63.66% and 66.79%, respectively. When the roasting time was 90 min, the iron grade decreased to 59.56%, and the iron recovery increased to 68.28%. The reason for this phenomenon is that with the extension of roasting time, the reduction reaction tends to be complete, the coke is exhausted, and part of the elemental iron is oxidized to FeO again, leading to a reduction of metallic iron and ferric tetroxide [21–23]. Therefore, considering economy and efficiency, a roasting time of 60 min is a good choice.

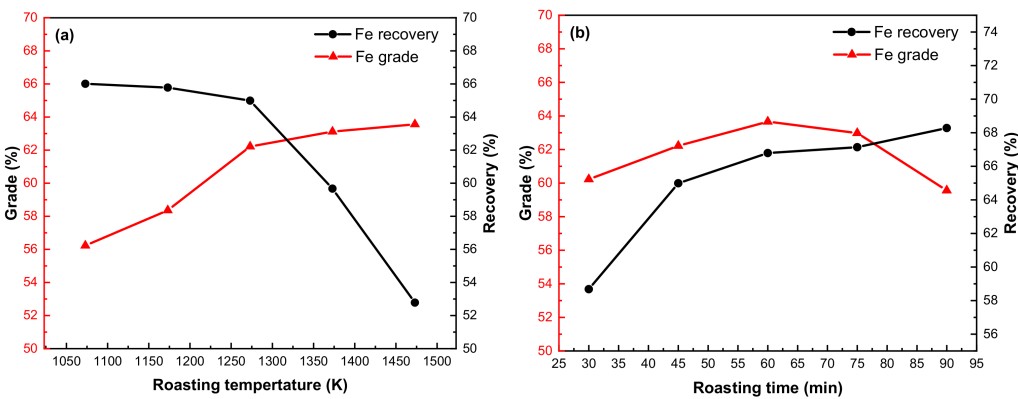

**Figure 3.** Effects of roasting temperature (**a**) and roasting time (**b**) on iron extraction.

### 3.3.3. Effect of Calcium Chloride Dosage

Tests with different calcium chloride dosages were carried out under the following conditions: roasting temperature of 1273 K, roasting time of 60 min, coke dosage of 8%, coke size of −1.5 mm, low-intensity magnetic field intensity of 0.1 T, and grinding fineness of −0.08 mm occupying 90%. It is known from Figure 4a that when the calcium chloride dosage was increased in the system of hydrogen chloride gas, the more conducive it was to improving the conversion rate of ferric chloride, but it also led to excess hydrogen chloride gas with other elements such as magnesium, aluminum, and manganese chloride. The amount of volatile chloride in the system does not favor the ferric chloride by hydrogen reduction into metallic iron, reducing the iron grade of iron concentrate [24,25]. When the amount of calcium chloride was increased to 10%, the iron grade of the iron concentrate increased to 66.25% and the iron recovery increased to 69.73%. Calcium chloride content increased to 12%, and iron grade and iron recovery decreased to 65.23% and 67.93%, respectively, under a calcium chloride dosage of 12%. This further indicates that a 10% calcium chloride dosage is a reasonable amount of chlorination agent.

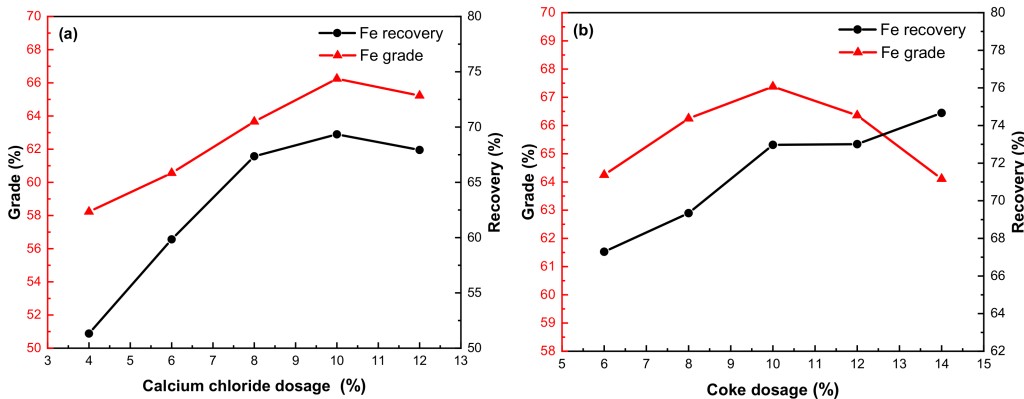

**Figure 4.** Effects of calcium chloride dosage (**a**) and coke dosage (**b**) on iron extraction.

### 3.3.4. Effect of Coke Dosage

Tests with different coke dosages were carried out under the following conditions: roasting temperature of 1273 K, roasting time of 60 min, calcium chloride dosage of 10%,

coke size of −1.5 mm, low-intensity magnetic field intensity of 0.1 T, and grinding fineness of −0.08 mm occupying 90%. The results in Figure 4b reveal that with the increase of coke consumption, the iron grade of the iron concentrate changed regularly at first and then decreased, and the iron recovery showed an increasing trend, which was related to the double role of coke in reducing gas CO and $H_2$ and adsorbing metal particles in the roasting process. Sufficient coke can provide sufficient CO and $H_2$ for the roasting process, which is conducive to improving the iron grade and iron recovery of iron concentrate. At the same time, a large amount of heat is released from the combustion of coke, and the rise in temperature promotes the reduction of iron oxide and iron chloride. These results further indicate that a coke dosage of 10% was a good choice, and iron concentrate with an iron grade of 67.38% and iron recovery of 72.97% was obtained.

### 3.3.5. Effect of Coke Size

Tests with different coke dosages were carried out under the following conditions: roasting temperature of 1273 K, roasting time of 60 min, calcium chloride dosage of 10%, coke dosage of 10%, low-intensity magnetic field intensity of 0.1 T, and grinding fineness of −0.08 mm occupying 90%. Figure 5a confirms that with the decrease of coke size, the specific surface area of coke increases and the adsorption capacity of coke is enhanced, which is conducive to the adsorption of metal particles on the coke surface and the improvement of iron recovery by low-intensity magnetic separation. The reduction of coke size is conducive to the improvement of iron grade of iron concentrate, but when the coke size is too small, the coke is easily consumed in a short time, and the chemical reaction has not been completed, which is not conducive to the recovery of iron [26,27]. Therefore, a coke size of −1 mm is more reasonable, and iron concentrate with an iron grade of 71.25% and iron recovery of 77.85% can be obtained.

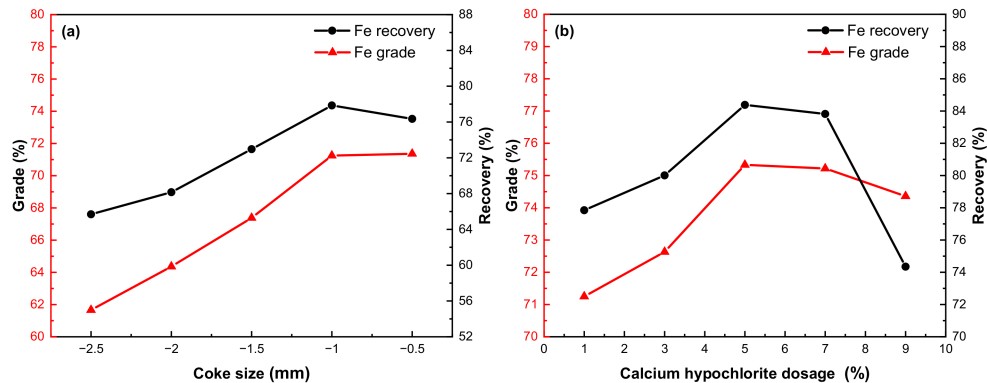

**Figure 5.** Effects of coke size (**a**) and calcium hypochlorite dosage (**b**) on iron extraction.

### 3.3.6. Effect of Additives Dosage

Adding proper additives in the roasting process can improve the index of concentrate products. A chemical reaction between calcium hypochlorite (an auxiliary) and hydrogen chloride in the system occurred, forming calcium chloride and chlorine [28–30]. Tests with different calcium hypochlorite dosages were carried out under the following conditions: roasting temperature of 1273 K, roasting time of 60 min, calcium chloride dosage of 10%, coke dosage of 10%, coke size of −1 mm, low-intensity magnetic field intensity of 0.1 T, and grinding fineness of −0.08 mm occupying 90%.

Adding calcium hypochlorite can significantly improve the iron grade of iron concentrate. With the increased dosage of calcium hypochlorite, the chlorine generated by the reaction accelerated the chlorination reaction of Fe to a certain extent, contributing to the formation of $FeCl_3$. In addition, excessive calcium hypochlorite may lead to excessive consumption of hydrogen chloride. On the other hand, hydrogen chloride involved in the chlorination reaction during the segregation roasting process is the most important chemical reaction. The results are shown in Figure 5b confirm that iron concentrate with an

iron grade of 75.33% and iron recovery of 84.38% was obtained under an optimal calcium hypochlorite dosage of 5%.

### 3.3.7. Grinding Fineness

Tests with different levels of grinding fineness were carried out under the following conditions: roasting temperature of 1273 K, roasting time of 60 min, calcium chloride dosage of 10%, calcium hypochlorite dosage of 5%, coke dosage of 10%, coke size of −1 mm, and low-intensity magnetic field intensity of 0.1 T. Figure 6a shows that with the increase of grinding fineness, an optimal increase of the iron grade in the iron concentrate can be observed. Due to the increased grinding fineness, the liberation of mineral monomer increased. However, excessively high grinding fineness will lead to the decrease of iron recovery in the iron concentrate, along with significantly increased grinding costs. Therefore, a grinding fineness of −0.06 mm occupying 90% was an ideal grinding parameter, and iron concentrate with an iron grade of 78.63% and iron recovery of 83.73% was obtained.

### 3.3.8. Low-Intensity Magnetic Field Intensity

Tests with different magnetic field intensities were carried out under the following conditions: roasting temperature of 1273 K, roasting time of 60 min, calcium chloride dosage of 10%, calcium hypochlorite dosage of 5%, coke dosage of 10%, coke size of −1 mm, and grinding fineness of −0.06 mm occupying 90%. The results in Figure 6b show that with the increased magnetic field intensity, a clear decrease in the iron grade and an increase in the iron recovery can be observed. With decreased magnetic field intensity, the materials will undergo a combined action by mechanical and magnetic forces during the magnetic separation process. Under magnetic field intensity from 0.06 T to 0.18 T, the iron grade of the iron concentrate decreased from 81.12% to 69.25%, while the iron recovery increased from 61.78% to 92.03%, which indicated that the increasing magnetic field intensity has a positive influence on the increase of overall iron recovery. However, it was not beneficial to improve the iron grade of the concentrate. A low-intensity magnetic separation field of 0.14 T was optimal, and an iron grade and iron recovery of iron concentrate were 75.32% and 92.02%, respectively.

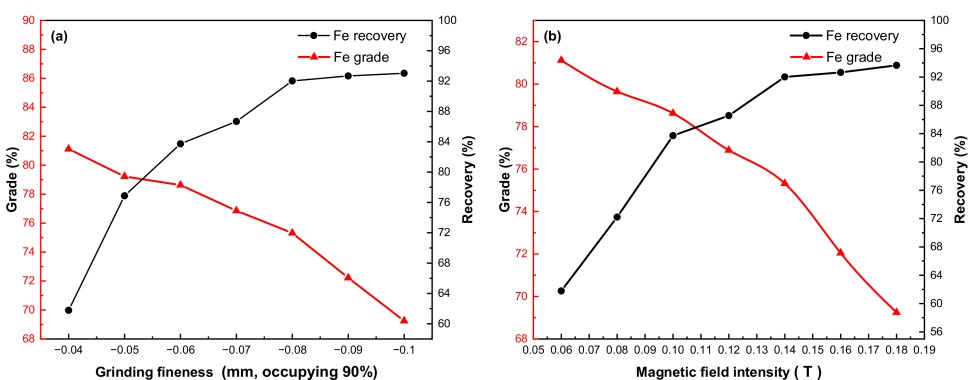

**Figure 6.** Effect of grinding fineness (**a**) and magnetic field intensity (**b**) on iron extraction.

### 3.4. Manganese Extraction from Low-Intensity Magnetic Separation Tailings

Iron was extracted from the coarse manganese concentrate by a roasting–low-intensity magnetic separation process. Since manganese mineral is a weakly magnetic mineral, it enters into tailings during the low-intensity magnetic separation process, and manganese is enriched from 28.65% to 32.25%. This further recovered manganese from the low-intensity magnetic separation tailings. These tests' conditions were roasting temperature of 1273 K, roasting time of 60 min, calcium chloride dosage of 10%, calcium hypochlorite dosage of 5%, coke dosage of 10%, coke size of −1 mm, and grinding fineness of −0.06 mm occupying 90%.

The results in Tables 3 and 4 show that with the increase of magnetic field intensity, the grade of manganese in manganese concentrate decreases and the recovery increases. The magnetic field intensity decreased from 0.45 T to 0.75 T, the manganese grade of manganese concentrate decreased from 55.63% to 38.84%, and the manganese recovery increased from 82.25% to 97.78%. After comprehensive consideration, the magnetic field intensity of 0.65 T is more appropriate for manganese extraction by high intensity magnetic separation, and the manganese concentrate with a manganese grade of 53.42% and manganese recovery of 94.49% was obtained.

**Table 3.** Main chemical composition analysis of low-intensity magnetic separation tailings (%).

| Fe | Mn | $Al_2O_3$ | $SiO_2$ | MgO | CaO | $Na_2O$ |
|---|---|---|---|---|---|---|
| 4.36 | 32.25 | 6.86 | 28.66 | 4.49 | 3.03 | 0.59 |

**Table 4.** Manganese extraction results from the low-intensity magnetic separation tailings (%).

| Magnetic Filed Intensity (T) | Products | Yield | Mn Grade | Mn Recovery |
|---|---|---|---|---|
| 0.45 | Mn concentrate | 47.68 | 55.63 | 82.25 |
| | Tailings | 52.32 | 10.94 | 17.25 |
| | Totals | 100.00 | 32.25 | 100.00 |
| 0.55 | Mn concentrate | 53.16 | 54.22 | 89.33 |
| | Tailings | 46.84 | 7.35 | 10.07 |
| | Totals | 100.00 | 32.27 | 100.00 |
| 0.65 | Mn concentrate | 57.29 | 53.42 | 94.49 |
| | Tailings | 42.71 | 3.88 | 5.51 |
| | Totals | 100.00 | 32.26 | 100.00 |
| 0.75 | Mn concentrate | 81.22 | 38.84 | 97.78 |
| | Tailings | 18.78 | 3.81 | 2.22 |
| | Totals | 100.00 | 32.26 | 100.00 |

*3.5. The Scale-Up Test of Iron and Manganese Extraction*

In order to further validate test repeatability and investigate the iron and manganese extraction index of the whole roasting–low-intensity magnetic separation–high-intensity magnetic separation process, repeated scale-up tests were carried out, and the optimal test conditions were show in Figure 7. These tests were scaled to a coarse manganese concentrate of 500 g for each roasting test, each test was repeated three times and the thrice-roasted ores were merged and mixed as magnetic separation materials. The results in Figure 7 and Table 5 show that the extraction index of iron and manganese was superior to the previous single conditional test, and further confirm that the test results are reliable and reproducible.

**Table 5.** Main chemical composition of iron concentrate and manganese concentrate (%).

| Products | Fe | Mn | S | P | $Al_2O_3$ | $SiO_2$ | MgO | CaO | $Na_2O$ |
|---|---|---|---|---|---|---|---|---|---|
| Fe concentrate | 78.63 | 0.86 | 0.06 | 0.05 | 1.22 | 6.88 | 1.66 | 4.22 | 0.23 |
| Mn concentrate | 4.12 | 53.55 | 0.08 | 0.04 | 0.98 | 14.86 | 2.87 | 6.23 | 0.25 |

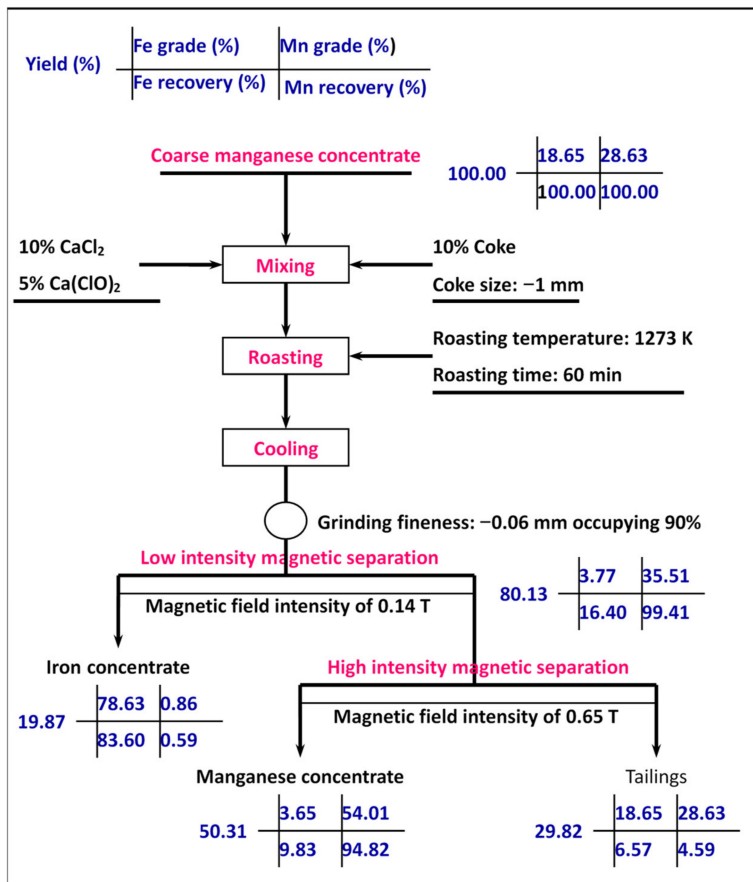

**Figure 7.** Scale-up test results of iron and manganese extraction from the coarse manganese concentrate (%).

### 3.6. Phase Transformation Mechanism in Roasting Process

An ideal extraction of manganese and iron was obtained from the coarse manganese concentrate by segregation roasting–low-intensity magnetic separation–high-intensity magnetic separation process in order to find the main minerals' transformation in segregation roasting. X-ray diffraction was used to analyze the main minerals' composition in iron concentrate and manganese concentrate. Scanning electron microscopy and energy dispersive spectroscopy (SEM–EDS) were used to analyze iron concentrate and manganese concentrate, respectively. The XRD and SEM–EDS analysis characterization results of the roasted ore and leaching residue are shown in Figures 8 and 9, respectively.

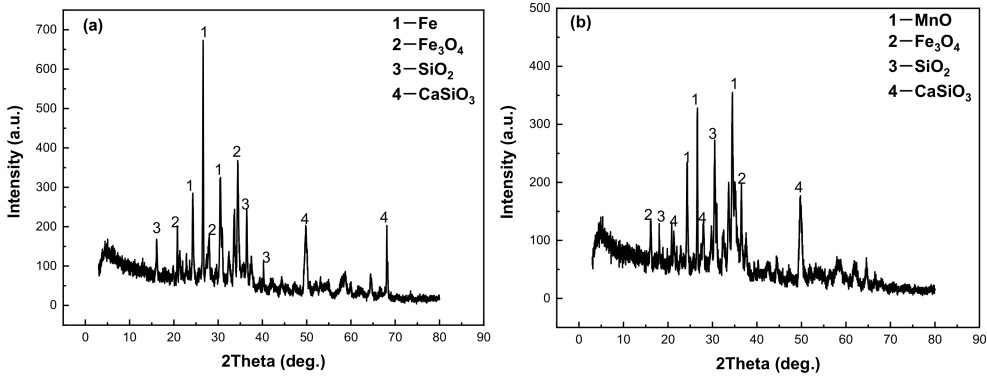

**Figure 8.** X-ray diffractogram of iron concentrate (**a**) and manganese concentrate (**b**).

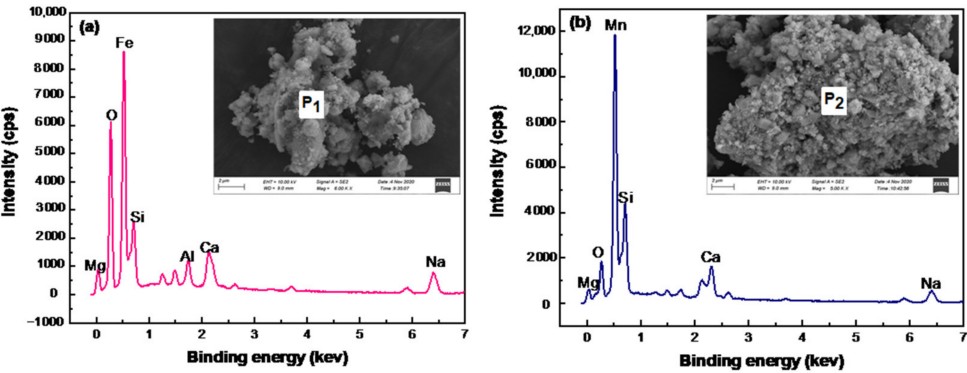

**Figure 9.** SEM–EDS images of iron concentrate (**a**) and manganese concentrate (**b**).

The course of roasting involves the chlorination reaction of iron oxide and the subsequent reduction reaction. During the heating process, rhodochrosite breaks down into manganese oxide (MnO) and carbon dioxide ($CO_2$), limonite loses crystalline water to form hematite, and the chlorinating agent reacts with water vapor and silica in the ore to generate hydrogen chloride [31–34]. Then, the hydrogen chloride reacts with metal oxides ($Fe_2O_3$) and manganese oxides (MnO) to generate $FeCl_3$ and $MnCl_2$, respectively. Finally, the $FeCl_3$ and $MnCl_2$ are reduced to metal iron and metal manganese on the surface of coke and the HCl is regenerated [35–39]. The iron phase transformation mechanism and manganese phase conversion during the roasting process are shown in Equations (4)–(15).

$$CaCl_2 + SiO_2 + H_2O_{(g)} = CaSiO_3 + 2HCl_{(g)} \tag{15}$$

$$Ca(ClO)_2 + 4HCl_{(g)} = 2Cl_{2(g)} + CaCl_2 + 2H_2O_{(g)} \tag{16}$$

$$C + H_2O_{(g)} = CO_{(g)} + H_{2(g)} \tag{17}$$

$$CO_{(g)} + 3Fe_2O_3 = 2Fe_3O_4 + CO_{2(g)} \tag{18}$$

$$3CO_{(g)} + Fe_2O_3 = 2Fe + 3CO_{2(g)} \tag{19}$$

$$CO_{(g)} + Fe_3O_4 = 3FeO + CO_{2(g)} \tag{20}$$

$$2FeO + Cl_{2(g)} + 4HCl_{(g)} = 2FeCl_{3(g)} + 2H_2O_{(g)} \tag{21}$$

$$1.5H_{2(g)} + FeCl_{3(g)} = Fe + 3HCl_{(g)} \tag{22}$$

$$MnCO_3 = MnO + CO_{2(g)} \tag{23}$$

$$MnO + 2HCl_{(g)} = MnCl_{2(g)} + H_2O_{(g)} \tag{24}$$

$$MnCl_{2(g)} + H_{2(g)} = Mn + 2HCl_{(g)} \tag{25}$$

The thermodynamic roasting results for Reactions (15)–(25) are shown in Figure 10, reflecting the relationship between standard Gibbs free energy ($\Delta G^\theta$) and temperature. As shown in Figure 9, when the temperature increases, the $\Delta G^\theta$ value of Reaction (15) gradually approaches negative values. When the temperature reaches 1273 K, the $\Delta G^\theta$ values of Reactions (15)–(22) are all negative, and these reactions can proceed spontaneously at this temperature. When the temperature increases from 673 K to 1473 K, the $\Delta G^\theta$ values of Reactions (24)–(25) are positive, and these reactions cannot proceed spontaneously at this temperature range. According to the Gibbs function, the more negative the Gibbs free energy changes, the higher the reaction tendency. When the temperature rises to 1273 K, the $\Delta G^\theta$ value of Reaction (16) is more negative than other reactions, which indicates adding calcium hypochlorite can promote the formation of ferric chloride. Therefore, the overall reaction sequence suggests that iron is easier to chlorinate and reduce than manganese oxide, and the reaction of manganese is mainly the decomposition reaction of rhodochrosite.

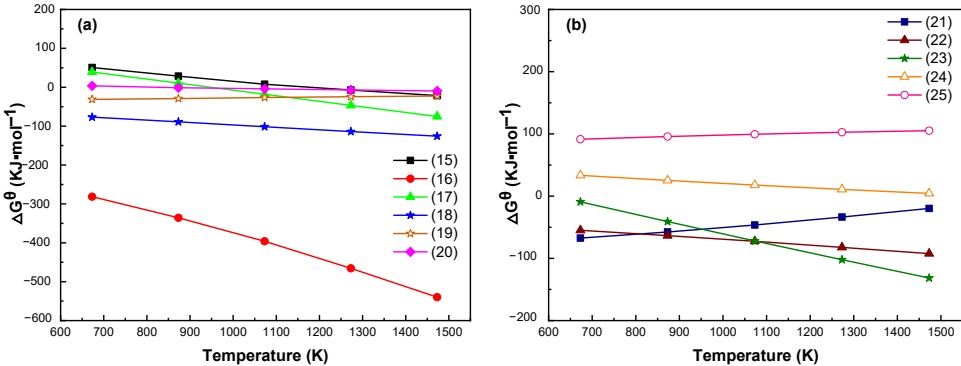

**Figure 10.** Correlation of standard Gibbs free energy (ΔGθ) with temperature for Reactions (**a**) Reactions (15)–(20); (**b**) Reactions (21)–(25).

## 4. Conclusions

Based on the results obtained in this work of iron and manganese extraction from the coarse manganese concentrate, we drew the following main conclusions:

(1) The coarse manganese concentrate contained 28.63% Mn and 18.65% Fe. Rhodochrosite, limonite, quartz, and olivine are the main minerals in coarse manganese concentrate, and the rhodochrosite and limonite have a close symbiotic relationship. Separation of manganese and iron was difficult to achieve with magnetic separation, gravity separation, or flotation.

(2) A novel process of segregation roasting with two stages of magnetic separation was used to treat the coarse manganese concentrate. The addition of calcium chloride and calcium hypochlorite synergistically enhanced the transformation of iron from a weakly magnetic mineral to a strongly magnetic mineral dominated by metallic iron and magnetite, and manganese changed from manganese carbonate to manganese oxide. Test results show that iron concentrate with an iron grade of 78.63% and iron recovery of 83.60%, and manganese concentrate with a manganese grade of 54.04% and manganese recovery of 94.82% were achieved under the following comprehensive conditions: roasting temperature of 1273 K, roasting time of 60 min, calcium chloride dosage of 10%, calcium hypochlorite dosage of 5%, coke dosage of 10%, coke size of −1 mm, grinding fineness of −0.06 mm occupying 90%, low-intensity magnetic field intensity of 0.14 T, and high-intensity magnetic field intensity of 0.65 T. Extraction of manganese and iron was straightforward.

(3) Phase transformation mechanism analysis results show that limonite ($Fe_2O_3 \cdot nH_2O$) was heated and lost crystalline water to form hematite ($Fe_2O_3$); hematite reacted with carbonic oxide (CO) and was reduced to magnetite in the reducing atmosphere. Some magnetite was reduced to ferrous oxide (FeO) by carbonic oxide (CO). Ferrous oxide reacted with hydrogen chloride and chlorine gas to form ferrous chloride. Ferrous chloride was reduced by carbon or hydrogen to become metallic iron (Fe) and was adsorbed onto the coke surface. Rhodochrosite ($MnCO_3$) is mainly a decomposition reaction to produce manganese oxide (MnO) in the roasting process. The thermodynamic calculation results and XRD and SEM–EDS analysis characterization results also further verified the phase transition mechanism of iron and manganese and the reliability of the test results.

**Author Contributions:** This is a joint work of the five authors; each author was in charge of their expertise and capability: J.X. for Writing, review, editing, and conceptualization, K.Z. for Software, T.C. for validation, W.X. for Data curation, and B.D. for Methodology. All authors have read and agreed to the published version of the manuscript.

**Funding:** This work was supported by the Sichuan Science and Technology Program (Grant Nos. 2021YJ0057, Nos. 2021YFG0268, Nos. 2019FS0451 and Nos. 2019FS0452); Funded by the Research Fund Program of Innovation Center of Rare Earth Resources Development and Utilization, China

**Institutional Review Board Statement:** Not applicable.

**Informed Consent Statement:** Not applicable.

**Data Availability Statement:** Not applicable.

**Conflicts of Interest:** The authors declare no conflict of interest. The authors alone are responsible for the content and writing of the article.

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
