# Peer review of "Extraction of Manganese and Iron from a Refractory Coarse Manganese Concentrate"

_metals, doi:10.3390/met11040563_

Round 1
Reviewer 1 Report
The article is written on an interesting and relevant topic. However, the text needs to be improved.
1. The research methods and the experimental design are insufficiently reflected.
2. Statistical and mathematical processing of the results is missing. The authors suggest optimal parameters, but there is no data about the design of experiments (DOE). Such data cannot be reliable.
3. Poor quality of graphical materials. Some dependencies are plotted on 4 points! Ranges are not justified. The text in figure 8 is unreadable
Author Response
Point 1: 1. The research methods and the experimental design are insufficiently reflected.
Response 1: We have perfected this part for your review. Thank you very much for your suggestions.
Point 2: 2. Statistical and mathematical processing of the results is missing. The authors suggest optimal parameters, but there is no data about the design of experiments (DOE). Such data cannot be reliable.
Response 2: Your suggestion is very helpful to our study, we used in the process of testing the condition of single factor experiment, does not take into account the orthogonal experiment, we try our best to added, please review again, thank you for your review of the work, we must strengthen the work of the follow-up research work, the experimental study on the work done as far as possible the reliability, ensure the accuracy of test results.
Point 3: 3. Poor quality of graphical materials. Some dependencies are plotted on 4 points! Ranges are not justified. The text in figure 8 is unreadable
Response 3: We have rechecked the test results. Due to the two process parameters of grinding fineness and magnetic field strength, the test data at different test points have a small gap, so there is some negligence in sorting out the data. According to your suggestion, we have supplemented the data of the test site, please review. Finally, Thank you very much for taking your precious time to review the article in your busy work, and thank you for your recognition of our research work. I hope our effort work can get your approval. Looking forward to your good news! With kind regards, Dr. Junhui Xiao
Reviewer 2 Report
Manuscript ID: metals-1138855
Title: Extraction of Manganese and Iron from a Refractory Rough Manganese Concentrate
Authors: Junhui Xiao et al.
Line 37-40, 41-44. Authors references for this information.
Line 50-85. Authors must give separate references for all six enrichment methods for manganese carbonate. Do not use “beneficiation”.
Line 86-87. Add full chemical formulas of minerals: rhodochrosite, magnesium rhodochrosite, kutnahorite, calcite, and siderite. Write chemical formulas to all minerals in the text in the first-time writing in brackets, for example – hematite (Fe2O3).
Line 103-118. Add references for manganese carbonate ore characterization.
Table 1. What is the LOI content? The sum of elements is not 100%, only 91.81wt. %.
Figure 1. Not all peaks are sign.
Line 209-225. What thermodynamic data and where did authors come from? Write values or provide a reference. Did authors used software: HSC Chemistry or FactSage?
Line 225. Add Boudouard reaction: 2CO = C + CO2
Figure 2. Add different colors for points and curves.
Line 353-356. Why authors used mixture of calcium chloride and calcium hypochlorite?
Table 4. Change Productivity to Yield.
Figure 7. Not all peaks are sign.
Figure 9. Add different colors for points and curves. Why is reaction (25) not added to figure 9b?
- The authors should add discussion about the future use of iron concentrate. It is necessary to add the content of sulfur and phosphorus, since if authors want to use this concentrate to make cast iron or steel, this is an important factor.
- Where does hydrogen gas (Н2) come from in the system: reactions 8, 13, 22 and 25? In the methods section authors do not write about this gas. Only about coal and additive.
- The technology described in the article assumes the release of gaseous HCl. What about the corrosion of the exhaust pipes of the units and the trapping of acid vapors? Authors should write about it.
- Authors must add a process flowsheet with metals balance (iron and manganese - yield, recovery, grade) and optimal process parameters.
References [1] and [2] are the same.
Author Response
Point 1: Line 37-40, 41-44. Authors references for this information.
Response 1: Thank you very much for your suggestions. We have reviewed relevant literature work and made supplements.
Point 2: Line 50-85. Authors must give separate references for all six enrichment methods for manganese carbonate. Do not use “beneficiation”.
Response 2: Thank you very much for your suggestions. We have consulted relevant literature work and supplemented relevant literature on the treatment technology of different types of manganese ores.
Point 3: Line 86-87. Add full chemical formulas of minerals: rhodochrosite, magnesium rhodochrosite, kutnahorite, calcite, and siderite. Write chemical formulas to all minerals in the text in the first-time writing in brackets, for example – hematite (Fe2O3).
Response 3: We did not express clearly here. Thank you very much for your suggestions. We will make corresponding modifications, please review.
Point 4: Line 103-118. Add references for manganese carbonate ore characterization.
Response 4: Thank you very much for your suggestions and we have supplemented relevant literature.
Point 5: Table 1. What is the LOI content? The sum of elements is not 100%, only 91.81wt. %.
Response 5: Here is the content of main chemical components, and a small amount of other element content has not been analyzed. Therefore, the element content in Table 1 is close to 100%, but the total is not 100%.
Point 6: Figure 1. Not all peaks are sign.
Response 6: Thank you very much for your review. We will supplement and improve it again.
Point 7: Line 209-225. What thermodynamic data and where did authors come from? Write values or provide a reference. Did authors used software: HSC Chemistry or FactSage?
Response 7: It is calculated by consulting literature and combining with HSC Chemistry software.Corresponding supplement has been made. Thank you again.
Point 8: Line 225. Add Boudouard reaction: 2CO = C + CO2
Response 8: Thank you very much for your advice. We added the chemical reaction formula and carried out thermodynamic calculation.
Point 9: Figure 2. Add different colors for points and curves.
Response 9: Here is our drawing is not clear enough and has been modified. Please review it.
Point 10: Line 353-356. Why authors used mixture of calcium chloride and calcium hypochlorite?
Response 10: On the basis of our previous research on iron increase and phosphorus reduction in high-phosphorus iron ore, important research results have been obtained. Adding the mixture of calcium chloride and calcium hypochlorite in the roasting process of high phosphorus iron ore can significantly improve the grade and recovery of iron concentrate. Because the iron in the rough concentrate of manganese in the form of hematite is similar to the iron in the form of hematite in the high phosphorus iron ore, we use calcium chloride and calcium hypochlorite as a mixture, and the ideal separation effect of iron and manganese was obtained.
Point 11: Table 4. Change Productivity to Yield.
Response 11: Thank you very much for your advice. We will make corresponding revisions.
Point 12: Figure 7. Not all peaks are sign.
Response 12: Here is our work not careful enough and has been revised and improved.
Point 13: Figure 9. Add different colors for points and curves. Why is reaction (25) not added to figure 9b?
Response 13: Thank you again for your advice. There is a mistake in the serial number of the chemical reaction formula here. We will correct it for your review.
Point 14: The authors should add discussion about the future use of iron concentrate. It is necessary to add the content of sulfur and phosphorus, since if authors want to use this concentrate to make cast iron or steel, this is an important factor. Where does hydrogen gas (Н2) come from in the system: reactions 8, 13, 22 and 25? In the methods section authors do not write about this gas. Only about coal and additive. The technology described in the article assumes the release of gaseous HCl. What about the corrosion of the exhaust pipes of the units and the trapping of acid vapors? Authors should write about it.
Response 14: This makes our analysis work rigorous. The content of sulfur and phosphorus in iron concentrate is an important factor affecting the product index. I rechecked the original test results and supplemented the content of sulfur and phosphorus in iron concentrate and manganese concentrate. In the process of roasting, chlorine salt and coke are added in this study, and the roasting temperature is high, the water in the material has become a gaseous state, and it is easy to react with coke by water gas to generate carbon monoxide (CO) and hydrogen (H2). For the generated hydrogen chloride gas, we are doing tail gas treatment research. The current research shows that hydrogen chloride is absorbed by limestone water to become calcium chloride, which can be used as chlorination agent in the roasting process after concentration, and the effect is consistent with the effect of adding pure calcium chloride. Again, thank you for your questions, which will provide important help for our follow-up research work. We also look forward to your valuable suggestions.
Point 15: Authors must add a process flowsheet with metals balance (iron and manganese - yield, recovery, grade) and optimal process parameters.
Response 15: Thank you for your suggestions and we have supplemented them to better reflect our research process.
Point 16: References [1] and [2] are the same.
Response 16: Thank you for your review. We have deleted duplicate references.
Finally, Thank you very much for taking your precious time to review the article in your busy work, and thank you for your recognition of our research work. I hope our effort work can get your approval. Looking forward to your good news!
Yours sincerely,
Dr. Junhui Xiao
Reviewer 3 Report
The manuscript is about the recovery of iron and manganese from rough concentrate manganese supplied by china's ore. The research topic is very interesting for the readers of the journal. The experimental activities are well described and logically planned. The novelty is clear and confirmed by the obtained results. Below a series of suggestion to improve the paper's quality:
- The authors should discuss the economic feasibility of the proposal process. A comparison with traditional recovery processes taking into account the Fe and Mn grade/recovery vs operating costs is strongly suggested.
- In the abstract, it should be emphasized that mine samples are used.
- How was determined the chemical composition of rough manganese concentrate? It was obtained by AAS? Which was the chemical attacks procedure?
- Some results are reported in section 'Materials and Methods'. I suggest moving them in results' section.
- Check Table 2, unit of measure is missing.
Author Response
Point 1: The authors should discuss the economic feasibility of the proposal process. A comparison with traditional recovery processes taking into account the Fe and Mn grade/recovery vs operating costs is strongly suggested.
Response 1: There are real economic issues here. We used this method to increase iron and reduce phosphorus in high phosphorus iron ore, and obtained ideal iron concentrate product index. Iron and manganese concentrates can be obtained by using this method here, and good economic benefits can be achieved according to the current market price.
Point 2: In the abstract, it should be emphasized that mine samples are used.
Response 2: There is indeed some improper expression here, we will revise it, please review it, and thank you again for your advice.
Point 3: How was determined the chemical composition of rough manganese concentrate? It was obtained by AAS? Which was the chemical attacks procedure?
Response 3: Here is the method of chemical analysis, chemical composition of coarse manganese concentrate was obtained by AAS, and the detection instrument is a Z–2000 atomic absorption spectrophotometer (Hitachi Co., Ltd. Tokyo, Japan)
Point 4: Some results are reported in section 'Materials and Methods'. I suggest moving them in results' section.
Response 4: The statement here is too general, we have made appropriate adjustment, thank you again.
Point 5: Check Table 2, unit of measure is missing.
Response 5: Here are the mistakes in our work, and we will make the corresponding supplement. Thank you for your suggestions.
Finally, Thank you very much for taking your precious time to review the article in your busy work, and thank you for your recognition of our research work. I hope our effort work can get your approval. Looking forward to your good news!
Kind regards,
Dr. Junhui Xiao
Round 2
Reviewer 1 Report
In each section, there are conclusions like
«Therefore, considering economy and efficiency, a roasting time of 60 min is a good choice.»
The statement is not justified. How was it determined? What indicator was used? Proof of this conclusion is required.
The situation is similar in other sections. Optimal values must be justified. What is the criterion for optimality? Without this information, the results of the work cannot be trusted. For example, going back to Figure 3, why 60? Not 59, not 61. A clear justification of the optimum selection criterion is required, as it is important for the work.
Author Response
Point 1: 1. In each section, there are conclusions like
«Therefore, considering economy and efficiency, a roasting time of 60 min is a good choice. »
The statement is not justified. How was it determined? What indicator was used? Proof of this conclusion is required.
The situation is similar in other sections. Optimal values must be justified. What is the criterion for optimality? Without this information, the results of the work cannot be trusted. For example, going back to Figure 3, why 60? Not 59, not 61. A clear justification of the optimum selection criterion is required, as it is important for the work.
Response 1: Hereby, we sincerely thank you for your valuable suggestions on our research work. This study is mainly to consider the separation of iron and manganese in rough manganese concentrate from technical solutions. In different process conditions, the single-factor research method is adopted to carry out the influence of different process conditions on the separation of iron and manganese. When carrying out iron extraction tests, the conclusions of each process condition do have similarities. Here we hope to express the comparative tests of other experimental points when each test condition changes in a certain interval. Here we also get scientific conclusions from a lot of research work. As for the selection of the best process conditions you proposed, in fact, it is mainly concerned about the layout of the test points. In the process of roasting, the change of the test points is often within the allowable error range, so that a relatively reasonable test conditions can be obtained, and the test results are relatively reliable. In addition, we also verified the feasibility of the chemical reaction from the perspective of thermodynamics, and also verified the reliability of the experimental results by means of analytical testing methods. We hope that our research work can be recognized by you. Of course, we also hope that you can put forward valuable suggestions again. We will continue to thanks for your precious suggestions.
Kind regards,
Dr. Junhui Xiao
Reviewer 2 Report
The authors answered all the questions and significantly improved the article. Figure 7 is a flow chart, very well done and allows to fully understand the research results. Analysis of roasting products for phosphorus and sulfur is also a very important addition. The article looks very good, this research will be of interest to a wide range of specialists.
In this form, article "Extraction of Manganese and Iron from a Refractory Coarse Manganese Concentrate" can be accepted into Metals.
I wish the authors success in their future research.
Author Response
Point 1: The authors answered all the questions and significantly improved the article. Figure 7 is a flow chart, very well done and allows to fully understand the research results. Analysis of roasting products for phosphorus and sulfur is also a very important addition. The article looks very good, this research will be of interest to a wide range of specialists.
In this form, article "Extraction of Manganese and Iron from a Refractory Coarse Manganese Concentrate" can be accepted into Metals.
I wish the authors success in their future research.
Response 1: Thank you very much for your recognition of our research work. Especially thank you for your important guidance for our research work, which is very helpful for our future research work. I am looking forward to the opportunity of learning and communicating next time.
With kind regards,
Dr. Junhui Xiao
Round 3
Reviewer 1 Report
I recommend that you justify the data more thoroughly in the future. Success in your research.